# Factors Associated with the Effectiveness of Regimens for the Treatment of Tuberculosis in Patients Coinfected with HIV/AIDS: Cohort 2015 to 2019

**DOI:** 10.3390/diagnostics13061181

**Published:** 2023-03-20

**Authors:** Natália Helena de Resende, Silvana Spíndola de Miranda, Adriano Max Moreira Reis, Cristiane Aparecida Menezes de Pádua, João Paulo Amaral Haddad, Paulo Vitor Rozario da Silva, Dirce Inês da Silva, Wânia da Silva Carvalho

**Affiliations:** 1College of Pharmacy, Universidade Federal de Minas Gerais, Belo Horizonte 31270-901, Brazil; 2College of Medicine, Universidade Federal de Minas Gerais, Belo Horizonte 30130-100, Brazil; 3School of Veterinary, Universidade Federal de Minas Gerais, Belo Horizonte 31270-901, Brazil; 4Hospital Foundation of the State of Minas Gerais/Eduardo de Menezes Hospital, Belo Horizonte 30622-020, Brazil

**Keywords:** tuberculosis, HIV, AIDS, coinfection, effectiveness, drug therapy

## Abstract

(1) Background: Infection with the Human Immunodeficiency Virus (HIV) is a significant challenge for tuberculosis (TB) control, with increasing mortality rates worldwide. Moreover, the loss to follow-up is very high, with low adherence to treatment, resulting in unfavorable endpoints. This study aimed to analyze the effectiveness of TB treatment in patients coinfected with HIV/AIDS and its associated factors. (2) Methods: Patients coinfected with TB and HIV/AIDS at a Reference Hospital for infectious diseases were followed up for a maximum of one year from the start of TB treatment until cure or censorship (death, abandonment, and transfer) from 2015 to 2019. The Cox proportional model was used to identify risk factors for effectiveness. (3) Results: Of the 244 patients included in the cohort, 58.2% (142/244) had no treatment effectiveness, 12.3% (30/244) died, and 11.1% (27/244) abandoned treatment. Viral suppression at the onset of TB treatment (HR = 1.961, CI = 1.123–3.422), previous use of Antiretroviral Therapy (HR = 1.676, CI = 1.060–2.651), new cases (HR = 2.407, CI = 1.197–3.501), not using illicit drugs (HR = 1.763, CI = 1.141–2.723), and using the basic TB regimen (HR = 1.864, CI = 1.084–3.205) were significant variables per the multivariate Cox regression analysis. (4) Conclusion: TB treatment for most TB patients coinfected with HIV/AIDS was not effective. This study identified that an undetectable viral load at the beginning of the disease, previous use of ART, not using illicit drugs and not having previously taken anti-TB treatment are factors associated with successful TB treatment.

## 1. Introduction

Infection with the Human Immunodeficiency Virus (HIV) is a significant challenge for tuberculosis (TB) control, with increasing mortality rates worldwide [1]. Until the coronavirus (COVID-19) pandemic, TB was the leading cause of death from a single infectious agent [2]. An estimated 1.4 million deaths among HIV-negative people and an additional 187,000 deaths among people living with HIV (PLHIV) were reported in 2021 [2]. Many advances in tuberculosis control in Brazil were achieved in the last few years, with a decreasing trend in mortality in Brazil, but with different patterns among regions, with an increase in the poorest regions of the country, such as the North and Northeast regions [3]. Despite the decreasing burden of these two diseases, they still make a significant contribution to mortality [4]. Brazil is among the 30 countries with the highest burden of TB and HIV/AIDS in the world [2].

The immunological impairment caused by HIV/AIDS increases the multiplication of *M. tuberculosis* and the degree of illness with TB [1]. The joint treatment of TB and HIV/AIDS (coinfected) reduces mortality, TB recurrence, and the transmission of both diseases in the community [5]. According to the 2020 World Health Organization (WHO) report, the success rate of treating new TB cases was 85% and 76% in patients coinfected with HIV/AIDS [6].

TB treatment-related complications involve adverse drug reactions (ADRs), classified as major and minor. The most significant ADRs, such as hepatotoxicity, nephrotoxicity, psychosis, hypersensitivity, thrombocytopenia, hematological alterations, and rhabdomyolysis, usually require the suspension of treatment and a change in the therapeutic regimen [7].

Besides safety-related outcomes, other reasons for switching regimens involve treatment effectiveness. Viral suppression must be maintained in the HIV/AIDS treatment, and the cure of the disease must be secured in the TB treatment [8,9,10].

To achieve this objective, policies for the control of TB and HIV/AIDS are needed, especially in countries with a high rate of both diseases [11,12]. Moreover, the loss to follow-up is very high, with low adherence to treatment, resulting in unfavorable endpoints [13].

Evidence-based planning and policies are required to achieve the goals of the End TB Strategy [3]. Thus, increasing knowledge about the effectiveness of the treatment for coinfected patients will guide actions to achieve favorable outcomes. Given the above, the present study aimed to analyze the effectiveness of TB treatment in coinfected patients and the factors associated with treatment success.

## 2. Methods

The study was conducted at the Eduardo de Menezes Hospital, of the Hospital Foundation of the State of Minas Gerais (FHEMIG), a reference hospital for TB, HIV/AIDS, and other infectious diseases. The hospital is linked to the Unified Health System and is located in a metropolis in southeastern Brazil. Eduardo de Menezes Hospital serves outpatients and inpatients. All patients were followed during the treatment of TB in the hospital and in the outpatient clinic. A non-concurrent longitudinal cohort study of coinfected patients was conducted.

This open cohort studied patients who started TB treatment between January 2015 and December 2018 and were followed up for one year. The study was approved by the Research Ethics Committee of the Federal University of Minas Gerais (UFMG) (CAAE: 23692713.3.0000.5149) and FHEMIG (CAAE:23692713.2.3001.5124) in 2014.

We included patients aged ≥18 years diagnosed with TB and HIV/AIDS who started TB treatment in January 2015. We excluded patients with incomplete medical records and who had a change in diagnosis during follow-up.

Patients were identified retrospectively through notifications from the TB Notifiable Diseases Information System sent to the hospital epidemiology center, and the medical records were reviewed to verify the HIV/AIDS test results.

The diagnosis of TB was made through the evaluation of a specialist physician who evaluated the signs and symptoms and the results of laboratory tests, such as the detection of acid-fast bacilli in sputum smears or the positive culture of a sputum sample or another sterile site.

The diagnosis of TB and HIV/AIDS was carried out through the evaluation of a specialist physician. For TB signs and symptoms, results of laboratory tests such as the detection of acid-fast bacilli in sputum smears or the positive culture of a sputum sample or other local sterile material, were evaluated. For HIV/AIDS, the diagnosis was made through laboratory tests and rapid tests which detect antibodies against HIV.

We reviewed the medical records to collect information related to the study using a data collection tool that included the following explanatory variables: sociodemographic characteristics, such as sex, age (<50 years or ≥50 years), marital status, education, place of residence and skin color; characteristics related to lifestyle such as the use of tobacco, alcohol, and illicit drugs; clinical features such as the clinical form of TB, time of HIV diagnosis, multimorbidity (defined as the existence of two or more non-infectious diseases), hospitalizations, viral suppression (result of undetectable viral load, <50 copies/mL); characteristics related to pharmacological treatment such as the history of use of ARV regimens, a drug therapy regimen for TB and HIV/AIDS, change in antiretroviral regimens, adverse reactions to TB and HIV/AIDS treatment. The definitions of the variables under the HIV/AIDS [14,15] and TB [7] clinical protocols are described below:-New cases: Patients who never underwent anti-TB treatment or did so for up to 30 days;-Types of TB regimens: Basic regimen with Rifampicin (R), Isoniazid (H), Pyrazinamide (Z), and Ethambutol (E) (two months RHZE and four months RH) or for Meningoencephalitis and Osteoarticular (two months RHZE and ten months RH) and special regimens with individualized drugs (streptomycin/amikacin/capreomycin, ofloxacin/levofloxacin, rifabutin, terizidone, linezolid, and ethionamide);-Types of antiretroviral therapy (ART) regimens: Formulation containing Efavirenz (EFV), Integrase Inhibitors (INI), or others.

The ART change was defined as replacing or including at least one drug in the therapy, considering the Ministry of Health protocols in force during the study period [14,15]. The response variable was the TB treatment outcome. The definition of outcomes per the WHO and the Brazilian Ministry of Health [7,16] is described below:-Cure: Defined for initially bacilliferous patients with two negative smears in one stage of the treatment and another at the end, or clinical or radiological improvement;-Treatment completed: Patients who completed TB treatment with no evidence of failure but whose culture and sputum results were unavailable in the last month of treatment or at other stages of treatment;-Under treatment: Patients who are using TB drugs and have not completed their treatment at the end of the one year;-Abandonment: Considered when patients discontinue treatment for thirty consecutive days or more;-Death: Death during treatment;-Transfer: Considered when patients started treatment and were transferred to another service. The outcome was not assessed when they were transferred;-Failure: When patients have persistent sputum positivity at the end of treatment. Patients who were strongly positive at the beginning of treatment and remained in this situation until the fourth month or those with initial positivity followed by negative and new positivity for two consecutive months, starting at the fourth month of treatment, are also classified as a failure;-Effectiveness (treatment success): The sum of patients cured and who completed TB treatment.

Outcomes were assessed while patients were followed up in the hospital for one year after starting TB treatment. In this study, we evaluated the success of TB treatment as the main outcome, which can be cure or treatment completed. These outcomes were evaluated through a survival analysis, in which the positive outcome and not death was considered.

### Statistical Analysis

The data collected in the review of medical records were entered into a database using the Microsoft Excel XP^®^ program and exported for analysis using the Statistical Analysis System (SAS). A descriptive analysis was performed to determine frequency, proportions, and median (interquartile range).

The sample size was calculated considering the several factors that interfere with the survival analysis, mainly the Hazard Ratio (HR) and the recruitment period [17]. The minimum calculated sample was 160 patients.

The Kaplan–Meier test was adopted to estimate the probability of treatment effectiveness. The log-rank test was used to compare treatment effectiveness per the explanatory variables. The association between treatment effectiveness and explanatory variables was verified using the Cox regression model. Variables with *p*-value ≤ 0.20 in the univariate analysis were included in the multivariate Cox regression analysis. The variables with *p* < 0.05 remained in the final Cox regression model. The magnitude of the association was measured by the Relative Hazard obtained by the Cox proportional hazards model.

## 3. Results

A total of 611 TB patients were reported during the study period, of which 407 were coinfected. Figure 1 shows the diagram of the inclusion of patients in the cohort.

In the total of 244 patients, the median age was 37 years (interquartile range = 16). Regarding sex, 78.7% were men with low schooling levels (61.9%) and undergoing TB treatment under the basic regimen (76.2%). Table 1 presents the sociodemographic and clinical characteristics of lifestyle and drug therapy.

In the cohort of 244 patients, the cure or TB treatment completion rate was 41.8%, the transfer rate 29.9%, the death rate 12.3%, and the abandonment rate 11.1%. Approximately 4.9% of patients were still under treatment at the end of the one-year follow-up period.

Comparing the time to cure according to the Log-Rank test of the Kaplan–Meier analysis shows a significant difference for sex, illicit drug, hospitalization, and viral suppression (Table 1). The time for healing for females and for patients who required hospitalization during TB follow-up is longer.

The Kaplan–Meier survival curves and the variables that remained significant in the final Cox model were illicit drug use (log-rank *p* = 0.014) and viral suppression at the onset of TB treatment (log-rank *p* = 0.012). The time for healing of those who do not use illicit drugs is shorter when compared to those who use them. In addition, the time to cure for those who achieved virological suppression at the beginning of TB treatment was shorter when compared to those who did not.

We observed that those who used the basic TB regimen, 82/186 (44.1%), healed more than those under a special regimen, 20/58 (34.5%). We identified that 186 (76.2%) patients used the basic regimen, and 46 (24.7%) had drug-induced hepatotoxicity.

The opportunistic infections identified were candidiasis (*n* = 197, 18.6%), pneumonia (*n* = 99, 9.4%), pneumocystosis (*n* = 74, 7.0%), syphilis (*n* = 54, 5.1%), toxoplasmosis (*n* = 35, 3.3), and herpes virus infection (*n* = 35, 3.3%). The most frequent non-infectious comorbidities were depression (*n* = 36, 3.4%) and arterial hypertension (*n* = 16, 1.5%).

Initial EFV-based ART was used in 155 patients, and 56 were on INI-containing regimens (raltegravir or dolutegravir). The HIV/AIDS treatment was modified in 81 patients.

ADRs were observed in 38.5% (32/81) of the patients, followed by therapeutic failure in 9.9% (8/81), which were the reasons for switching medications. There was no significant difference in TB treatment outcome among patients who switched ART at follow-up (log-rank *p* = 0.6679).

The drugs causing ADRs were RHZE (*n* = 53, 31.9%), sulfamethoxazole + trimethoprim (*n* = 29, 17.5), and tenofovir (*n* = 11, 6.6%). The ADRs were hepatotoxicity (*n* = 46, 15.49%), acute renal failure (*n* = 30, 10.10%), and skin reactions (*n* = 20, 6.73%).

Univariate Cox regression analysis identified an association with time to cure *p* < 0.05 for sex (HR = 2.133, CI = 1.109–4.100), illicit drug (HR = 1.663, CI = 1.097–2.522), and hospitalization (HR = 1.647, CI = 0.989–2.744).

Viral suppression at the onset of TB treatment (HR = 1.961, CI = 1.123–3.422), previous use of ART (HR = 1.676, CI = 1.060–2.651), TB new cases (HR = 2.407, CI = 1.197–3.501), not using illicit drugs (HR = 1.763, CI = 1.141–2.723), and using the basic TB regimen (HR = 1.864, CI = 1.084–3.205) were significant variables according to the multivariate Cox regression analysis (Table 2).

## 4. Discussion

Achieving cure in TB patients coinfected with HIV/AIDS is a significant challenge, as it involves behavioral aspects related to the immune system and treatment. In this respect, this study identified that an undetectable viral load at the beginning of the disease, the previous use of ART, not using illicit drugs, and not having been previously submitted to anti-TB treatment are associated with a higher risk of treatment success. As described by other authors who evaluated the outcome of TB in coinfection, late diagnosis of HIV/AIDS and delayed onset of TB treatment negatively interfere with treatment success [18,19]. Therefore, the recommendation to test all patients with TB for HIV/AIDS and evaluate all patients with HIV/AIDS for TB should be followed [2].

Having an undetectable viral load does not prevent the individual from developing TB but interferes with the risk of achieving a favorable outcome. Late TB diagnosis in patients with HIV/AIDS, or vice versa, prevents early treatment and contributes to the spreading of the bacillus in the community and higher mortality [20]. Thus, starting treatment as early as possible for these patients is essential.

According to the WHO, global HIV testing coverage among people diagnosed with TB remained high in 2020 (73%). However, the absolute number of people with TB who knew their HIV status dropped from 4.8 million in 2019 to 4.2 million in 2020, a 15% reduction [21]. Brazil also recorded a decrease in new TB cases tested for HIV, 82.2% in 2020 and 76.9% in 2021 [22].

Viral suppression and the use of ART interfere with the survival of coinfected individuals, as observed in a previous study carried out in Ethiopia [11]. Therefore, ART should be initiated regardless of the CD4 T-lymphocyte count [15].

The lack of a significant difference in the survival analysis comparing different ARTs identified in this study agrees with previous investigations, especially when evaluating the comparison between INI and EFV [23,24,25].

ART programs in several countries, including Brazil, shifted from using non-nucleoside analog reverse transcriptase inhibitors, such as EFV, to INI during the period in which the cohort was formed. INIs have superior efficacy, greater tolerability, and a high genetic barrier [26]. However, concerns arose about the compromise in the therapeutic response of these drugs due to the drug–drug interaction of INI with rifampicin, requiring an increase in the dose in coinfected patients. Despite this disadvantage vis-à-vis EFV and the greater complexity of the treatment regimen, real-world studies have shown that dolutegravir does not negatively impact viral suppression [26]. Another study that evaluated raltegravir in coinfection indicated that the use of the medication twice a day compromises adherence and, consequently, the effectiveness of the treatment, but the alternative use of this medication is recommended [27].

Adherence is essential, primarily when referring to long treatments with complex regimens that must be used more than once daily. The posologic convenience attributed to the tablets available in a fixed dose combined as in the association (RHZE) for TB and HIV/AIDS (Tenofovir difumarate, Lamivudine, and EFV) may favor treatment adherence and, consequently, effectiveness [28,29].

During follow-up, it was observed that this population is difficult to monitor, considering the high rate of abandonment, death, and transfer. Those are administrative causes and need to be better elucidated through the development of public policies and new qualitative studies that assess the social issues in which affect patients, such as the use of illicit drugs and the experience with the use of multiple medications in coinfection [30].

The abandonment rate in this study was high and higher than reported in other studies [31]. The outcome of abandonment occurred in 11.1% of cases, a finding that should be analyzed carefully, since Brazil is among the 30 countries with the highest burden of TB and HIV/AIDS in the world [2]. Abandonment can contribute to multidrug resistance, further aggravating TB control in Brazil [2].

Depression and hypertension were the most frequent non-infectious diseases, as observed in other studies that evaluated comorbidities in patients with HIV/AIDS [31,32], which raises the need to monitor the risk of cardiovascular diseases in this population as well as understand patients’ feelings about their treatment in order to respond to the foundation of these diseases. These diseases may be related to the environment in which individuals live or their behavior. Depression affects quality of life [33,34,35], and the integration of mental health services with TB/HIV programs can facilitate the identification and timely treatment of common mental disorders. There is an association between quality of life and ADR, educational level, and vulnerability [36]. Although the use of tobacco and alcohol are not statistically significant, Mycobacterium tuberculosis is more likely to develop in these individuals and influences the prognosis and treatment of the disease [37], in addition to being highly prevalent in PLHIV [38].

In this cohort, not using illicit drugs was a significant variable for curing or ending TB treatment. Several studies that evaluated the TB outcome identified drug addiction as a predictor of noncompliance [39,40]. Early drug use identification can lead to treatment adherence if the health team is adequately trained to identify and implement the directly observed treatment in this population [41]. Moreover, new cases were the most cured patients, since individuals with a noncompliance history are more likely to re-interrupt treatment or have poor adherence [42].

As observed, the presence of other associated diseases in addition to coinfection can worsen the clinical situation of these patients, who are in advanced immunodeficiency, since at the beginning of treatment, 70.9% of the patients had a CD4 T lymphocyte count below 200 cells/mm^3^ [15]. This favors the development of opportunistic infections and the need of prophylaxis, revealing greater polypharmacy in these individuals [43].

The complexity of drug therapy associated with the need to treat other chronic diseases and opportunistic infections still favors the emergence of ADRs. ADRs were the cause of ART and TB treatment changes, mainly due to hepatotoxicity and acute renal failure. These data agree with other studies that evaluated ADRs in coinfected patients [44,45].

The profile of these patients is in accordance with that described in the WHO report [2] and with studies carried out in Brazil [37], considering that they are mostly male patients, in the economically active age group, with low education, single marital status, and non-white majority. In this study, the time for healing was longer for females. When compared to the national average, the proportion of homeless patients is well beyond what is observed in the country. Among the 8070 cases of TB-HIV co-infection identified in 2019 in Brazil, 5.6% were homeless, and 5.9% were deprived of their liberty. In this study, 14.8% were homeless [46]. This is due to the fact that the hospital where the study was carried out is a reference hospital for infectious diseases, which receives the most critically ill patients residing mostly in the metropolitan region of Belo Horizonte. It was observed that during the follow-up period, 92% of the evaluated patients required hospitalization, indicating their severe health conditions and the need for longer follow-up time until successful TB treatment.

This population deserves special monitoring, since their social conditions can contribute to the abandonment of treatments and consequently make it difficult to control both diseases. Despite Brazil not being among the countries with the highest rate of multidrug-resistant TB, the abandonment can contribute to achieving this condition [2]. In addition, due to the high rate of abandonment, public policies that involve the inclusion of this population must be increased [47]. Partnership with social movements and interaction with other sectors that work in promoting human rights, racial equality, combating the abuse of licit drugs (such as tobacco and alcohol), and illicit are necessary [3].

Some limitations were observed in this study, such as the analysis of patients in only one reference hospital for infectious diseases. The inclusion of drug resistance TB patients in the study represents a limitation, but there were only four patients. Another factor is the use of secondary data, as there may be variables such as other clinical conditions, aspects related to lifestyle, and social factors that affected the outcome but were not documented in the medical records, besides transfers for which it was not possible to assess the outcomes. However, knowing the profile of these individuals and the risk factors involved in a successful treatment is crucial to improve TB management.

## 5. Conclusions

TB treatment for most TB patients coinfected with HIV/AIDS was not effective. This study identified that an undetectable viral load at the beginning of the disease, previous use of ART, not using illicit drugs, and not having previously taken anti-TB treatment are factors associated with successful TB treatment.

## Figures and Tables

**Figure 1 diagnostics-13-01181-f001:**
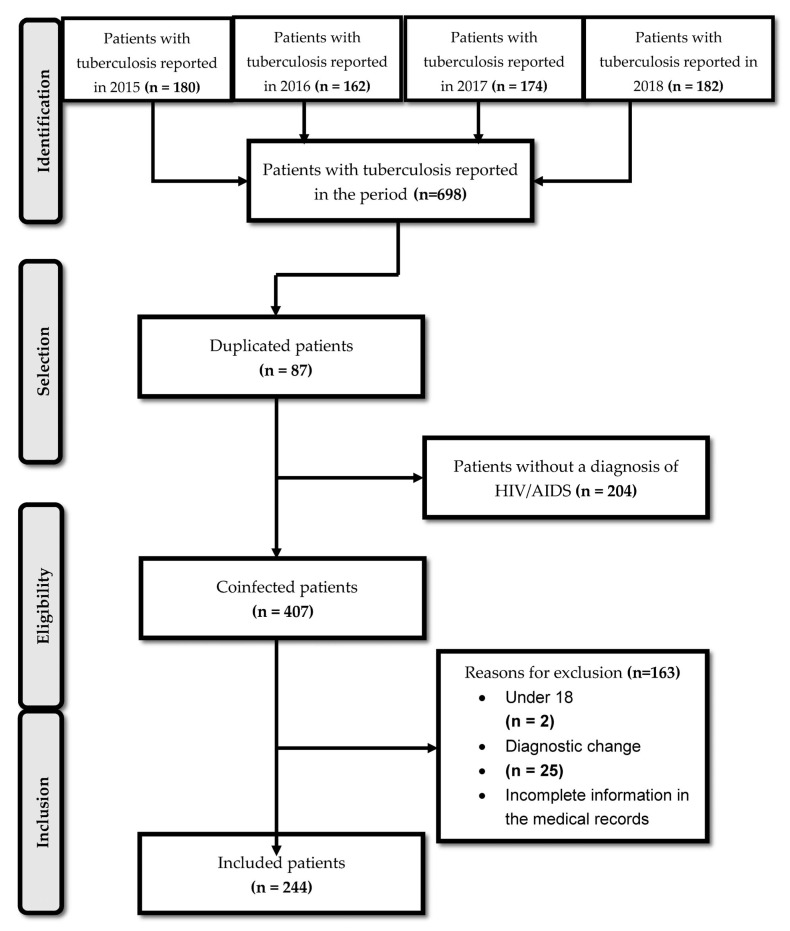
Inclusion and exclusion diagram of cohort patients.

**Table 1 diagnostics-13-01181-t001:** Sociodemographic, behavioral, clinical, and drug treatment characteristics (*n* = 244).

Variable	N	%	Treatment Success	%	Log-Rank *p*
Sex					
Male	192	78.69	92	47.92	0.0184
Female	52	21.31	10	19.23	
Age					
<50 years	200	81.97	82	41.00	0.1797
≥50 years	44	18.03	20	45.45	
Schooling					
≤8 years	151	61.88	58	38.41	0.1019
>8 years	75	30.74	38	50.67	
Marital status					
Single/Widower	194	79.51	85	43.81	0.6698
Married/Common-law marriage	49	20.08	16	32.65	
Skin color					
White	44	18.03	21	47.73	0.9490
Non-white	199	81.56	81	40.70	
Place of residence					
Metropolitan Region of Belo Horizonte	207	84.84	89	42.99	0.7633
Other Municipalities	37	15.16	13	35.14	
Special Status					
Deprived of liberty	13	05.33	7	53.85	0.1758
Living on the street	36	14.75	9	25.00	
None	195	79.92	86	44.10	
Tobacco use					
Yes	131	53.69	58	44.27	0.1328
No	96	39.34	39	40.63	
Alcohol use					
Yes	149	61.06	67	44.96	0.3776
No	82	33.61	31	37.81	
Illicit drugs					
Yes	107	43.85	47	43.93	0.0144
No	115	47.13	45	39.13	
TB Clinical Form					
Pulmonary and Extrapulmonary	64	26.23	35	54.69	0.4054
Pulmonary	118	48.36	42	35.59	
Extrapulmonary	62	25.41	25	40.32	
New TB case					
Yes	198	81.15	83	41.92	0.1479
No	43	17.62	16	37.21	
Multimorbidity					
Yes	29	11.89	12	41.38	0.8598
No	215	88.11	90	41.86	
Hospitalization					
Yes	225	92.21	84	37.33	0.0495
No	19	07.79	18	94.73	
HIV identification					
Before TB diagnosis	218	89.34	87	39.91	0.5453
After TB diagnosis	4	01.64	4	100	
Simultaneous diagnostics	21	06.61	11	52.38	
HIV diagnosis time					
<12 months	145	59.42	67	46.21	0.7934
≥12 months	95	38.93	35	36.84	
Baseline CD4 ≥ 200 cells/mm^3^					
No	161	70.93	66	40.99	0.1617
Yes	66	29.07	35	53.03	
HIV viral suppression					
No	193	85.02	80	41.45	0.0120
Yes	34	14.98	22	64.71	
History of ARV Use					
Yes	121	49.59	51	42.15	0.0856
No	118	48.36	51	43.22	
HIV/AIDS Treatment Regimens					
None	13	5.32	0	0	0.9961
Integrase Inhibitor	56	22.95	27	48.21	
Efavirenz	155	63.52	68	43.87	
Other	20	8.20	7	35.00	
TB Treatment Regimen					
Basic	186	76.22	82	44.10	0.0692
Special	58	23.77	20	34.48	
ADR TB/HIV					
No	160	65.57	64	40.00	0.4324
Yes	84	34.43	38	45.24	
ARV regimen change					
Yes	81	33.20	43	53.10	0.6679
No	163	66.80	59	36.20	

Abbreviations: ADR = Adverse Drug Reaction; ARV = antiretroviral; HIV = Human Immunodeficiency Virus; TB = tuberculosis.

**Table 2 diagnostics-13-01181-t002:** Cox regression analysis for risk factors for curing tuberculosis (*n* = 244).

	Univariate				Multivariate			
Variable	HR	95% CI		*p*	HR	95% CI		*p*
**Sociodemographic**								
Sex								
Female	1.000	-	-	-	-	-	-	-
Male	2.133	1.109	4.100	0.0231	-	-	-	-
Age								
<50 years	1.000	-	-	-	-	-	-	-
≥50 years	1.391	0.852	2.270	0.1869	-	-	-	-
Schooling								
≤8 years	1.000	-	-	-	-	-	-	-
>8 years	1.402	0.929	2.115	0.1080	-	-	-	-
Special Status								
None	1.000	-	-	-	-	-	-	-
Deprived of liberty	0.739	0.341	1.600	0.4425	-	-	-	-
Living on the street	0.550	0.277	1.094	0.0885	-	-	-	-
Behavioral and lifestyle								
Tobacco use								
No	1.000	-	-	-	-	-	-	-
Yes	0.732	0.485	1.106	0.1339	-	-	-	-
Illicit drugs								
Yes	1.000	-	-	-	-	-	-	-
No	1.663	1.097	2.522	0.0167	1.763	1.141	2.723	0.0106
Clinical								
New TB case								
No	1.000	-	-	-	-	-	-	-
Yes	1.477	0.863	2.526	0.1545	2.407	1.306	4.433	0.0048
Hospitalization								
Yes	1.000	-	-	-	-	-	-	-
No	1.647	0.989	2.744	0.0554	-	-	-	-
Baseline CD4 ≥ 200 cells/mm^3^								
Yes	1.000	-	-	-	-	-	-	-
No	0.749	0.497	1.130	0.1687				
Viral suppression								
No	1.000	-	-	-	-	-	-	-
Yes	1.812	1.125	2.917	0.0145	1.961	1.123	3.422	0.0178
Drug therapy								
History of ARV Use								
No	1.000	-	-	-	-	-	-	-
Yes	1.402	0.947	2.074	0.0912	1.676	1.060	2.651	0.0270
TB treatment								
Special	1.000	-	-	-	-	-	-	-
Basic	1.563	0.956	2.556	0.0748	1.864	1.084	3.205	0.0244

Abbreviations: ARV = antiretroviral; TB = tuberculosis.

## Data Availability

Data were obtained from the SUS of Belo Horizonte and are available from the coordinator researcher.

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
