# Peer review of "Factors Associated with the Effectiveness of Regimens for the Treatment of Tuberculosis in Patients Coinfected with HIV/AIDS: Cohort 2015 to 2019"

_diagnostics, 2023, doi:10.3390/diagnostics13061181_

Round 1
Reviewer 1 Report
The authors describe the factors affecting treatment success among TB/ HIV coinfected persons. The paper is well written, and the flow makes it easy to follow. The method section is well described the presentation of the results is in line with the analysis carried out.
A few suggestions below.
In the methods section, it is not clear what, if any exploration for interaction was carried out. A couple of final determinants of treatment success in this study may be related e.g. previous ARV use and VL suppression, or whether the new TB case gets the basic TB regimen. This could be explained further.
Additionally, the discussion section is quite comprehensive. It would help to start with the significant determinants of treatment success. e.g. illicit drug use line 274-278 could come before the discussion on adherence that is not measured in the study.
Finally, it would be interesting to see what research questions the authors would propose in light of these results e.g. qualitative studies among persons with TB who were drug users or in view of the study limitations identified.
Else, this was an excellent read and well done!
Author Response
The incorporated revisions are listed below.
Reviewer 1
Open Review
(x) I would not like to sign my review report
( ) I would like to sign my review report
English language and style
( ) English very difficult to understand/incomprehensible
( ) Extensive editing of English language and style required
( ) Moderate English changes required
(x) English language and style are fine/minor spell check required
( ) I don't feel qualified to judge about the English language and style
Yes |
Can be improved |
Must be improved |
Not applicable |
|
Does the introduction provide sufficient background and include all relevant references? |
(x) |
( ) |
( ) |
( ) |
Are all the cited references relevant to the research? |
(x) |
( ) |
( ) |
( ) |
Is the research design appropriate? |
(x) |
( ) |
( ) |
( ) |
Are the methods adequately described? |
( ) |
(x) |
( ) |
( ) |
Are the results clearly presented? |
(x) |
( ) |
( ) |
( ) |
Are the conclusions supported by the results? |
( ) |
(x) |
( ) |
( ) |
Comments and Suggestions for Authors
The authors describe the factors affecting treatment success among TB/ HIV coinfected persons. The paper is well written, and the flow makes it easy to follow. The method section is well described the presentation of the results is in line with the analysis carried out.
A few suggestions below.
In the methods section, it is not clear what, if any exploration for interaction was carried out. A couple of final determinants of treatment success in this study may be related e.g. previous ARV use and VL suppression, or whether the new TB case gets the basic TB regimen. This could be explained further.
Exploration was not carried for the interaction. Although patients with new TB cases started treatment with the basic regimen, during follow-up, we observed changes in medications for special regimens. So, these variables are different and address different characteristics. As well as the viral load at the beginning of the TB treatment and the previous use of ART, considering that the patient may have abandoned the treatment or that it may not have been effective.
Additionally, the discussion section is quite comprehensive. It would help to start with the significant determinants of treatment success. e.g. illicit drug use line 274-278 could come before the discussion on adherence that is not measured in the study.
After the suggestion of all the reviewers and the addition of some excerpts in the text, we chose to maintain the order of the paragraphs so that it has better coherence in the text. Although adherence was not measured it is related to the effectiveness of antiTB and antiretroviral treatments.
Finally, it would be interesting to see what research questions the authors would propose in light of these results e.g. qualitative studies among persons with TB who were drug users or in view of the study limitations identified.
Considering the results of studies with low frequency of treatment success during follow-up it was observed that this population is difficult to be followed, considering the high rate of abandonment, death and transfer. Those are administrative causes and need to be better elucidated through the development of public policies and new qualitative studies that assess the social issues in which patients are inserted, such as the use of illicit drugs and the experience with the use of multiple medications in co-infection. This excerpt was added to the text as suggested with reference to a qualitative study carried out in Brazil by our research group. Available at: https://www.mdpi.com/1660-4601/19/22/15153
Else, this was an excellent read and well done!

Reviewer 2 Report
This study addresses an important question; identification of factors that correlate with the effectiveness of TB treatment and treatment success in TB-HIV coinfected patients. The study was well designed, with a total of 244 dually infected patients included, and many variables were examined.
I found the discussion and analysis of results to be unclear. This study concluded that an undetectable viral load at the beginning of the disease, previous use of ART, not using illicit drugs, not having been previously submitted to anti-TB treatment, and using the basic TB regimen are associated with greater survival (Discussion, page 9). This is all intuitive, and consistent with other reports. However, Fig 2 appears to indicate that the usage of illicit drugs (red line, Fig. 2A) correlated with better survival estimates, and that virological suppression (blue line) correlated with shorter survival times. Furthermore, the study concludes that “Viral suppression and the use of ART interfere with the survival of coinfected individuals, as observed in a previous study carried out in Ethiopia (page 10, lines 235-6).” This is inconsistent with the other conclusions of the study, and with the previous study in Ethiopia cited that actually showed that viral suppression correlated with better survival.
The data summarized in Table 2 presents Hazard Ratios and 95% Confidence Intervals and P values for most variables, with higher HR values correlating with better clinical outcomes. Admittedly this reviewer is not a statistician, so I can’t evaluate the validity of the statistical analysis and whether this study was sufficiently powered for the number of variables examined. In general, it would be good to describe these results in a clearer, less ambiguous fashion.
Author Response
Reviewer 2
Open Review
(x) I would not like to sign my review report
( ) I would like to sign my review report
English language and style
( ) English very difficult to understand/incomprehensible
( ) Extensive editing of English language and style required
(x) Moderate English changes required
( ) English language and style are fine/minor spell check required
( ) I don't feel qualified to judge about the English language and style
Yes |
Can be improved |
Must be improved |
Not applicable |
|
Does the introduction provide sufficient background and include all relevant references? |
(x) |
( ) |
( ) |
( ) |
Are all the cited references relevant to the research? |
(x) |
( ) |
( ) |
( ) |
Is the research design appropriate? |
(x) |
( ) |
( ) |
( ) |
Are the methods adequately described? |
(x) |
( ) |
( ) |
( ) |
Are the results clearly presented? |
( ) |
( ) |
(x) |
( ) |
Are the conclusions supported by the results? |
( ) |
(x) |
( ) |
( ) |
Comments and Suggestions for Authors
This study addresses an important question; identification of factors that correlate with the effectiveness of TB treatment and treatment success in TB-HIV coinfected patients. The study was well designed, with a total of 244 dually infected patients included, and many variables were examined.
I found the discussion and analysis of results to be unclear. This study concluded that an undetectable viral load at the beginning of the disease, previous use of ART, not using illicit drugs, not having been previously submitted to anti-TB treatment, and using the basic TB regimen are associated with greater survival (Discussion, page 9). This is all intuitive, and consistent with other reports. However, Fig 2 appears to indicate that the usage of illicit drugs (red line, Fig. 2A) correlated with better survival estimates, and that virological suppression (blue line) correlated with shorter survival times. Furthermore, the study concludes that “Viral suppression and the use of ART interfere with the survival of coinfected individuals, as observed in a previous study carried out in Ethiopia (page 10, lines 235-6).” This is inconsistent with the other conclusions of the study, and with the previous study in Ethiopia cited that actually showed that viral suppression correlated with better survival.
In this study, we evaluated the success of TB treatment as the main outcome, which can be cure or end of treatment. Those outcomes were evaluated through a survival analysis, in which the positive outcome and not death was considered.
Figure 2 shows a survival curve in which the event of interest is the cure of TB. The time for healing of those who do not use illicit drugs is shorter when compared to those who use them. And the time to cure for those who achieved virological suppression at the beginning of TB treatment was shorter when compared to those who did not. For the avoidance of doubt we removed the figure and put the explanation in the text.
In the case of the study in Ethiopia, death was evaluated as the main outcome, and in our study we evaluated the success of TB treatment as the event that marks the end of the observation period, which represents a difference in the interpretation of the curve. In the survival analysis, the time to be analyzed is not restricted to the time until death, but until the occurrence of any event of interest. This part of the methods has been rewritten for clarity.
The data summarized in Table 2 presents Hazard Ratios and 95% Confidence Intervals and P values for most variables, with higher HR values correlating with better clinical outcomes. Admittedly this reviewer is not a statistician, so I can’t evaluate the validity of the statistical analysis and whether this study was sufficiently powered for the number of variables examined. In general, it would be good to describe these results in a clearer, less ambiguous fashion.
The results were described more clearly in the text, explaining that the main outcome evaluated was the cure of TB and therefore the Hazard Ratio values ​​are higher in better clinical results as suggested.

Reviewer 3 Report
This is a well-written paper highlighting the challenges and complexities in managing TB-HIV coinfection.
The analysis is comprehensive and well-conducted, identifying a whole range of factors that contribute to poorer outcomes.
Would be good to know if authors adjusted or accounted for other comorbidities such as diabetes, end stage renal failure which could also affect outcome.
The authors also identify many different factors e.g. social, behavioural, etc which contribute to poorer outcomes. I am sure that these are not specific to Brazil alone. The authors could further strengthen the paper by proposing how they could integrate these key changes into their TB programme, rather than targeting each factor individually, which may be challenging or slow to do.
Author Response
Reviewer 3
Open Review
( ) I would not like to sign my review report
(x) I would like to sign my review report
English language and style
( ) English very difficult to understand/incomprehensible
( ) Extensive editing of English language and style required
( ) Moderate English changes required
(x) English language and style are fine/minor spell check required
( ) I don't feel qualified to judge about the English language and style
Yes |
Can be improved |
Must be improved |
Not applicable |
|
Does the introduction provide sufficient background and include all relevant references? |
(x) |
( ) |
( ) |
( ) |
Are all the cited references relevant to the research? |
(x) |
( ) |
( ) |
( ) |
Is the research design appropriate? |
(x) |
( ) |
( ) |
( ) |
Are the methods adequately described? |
(x) |
( ) |
( ) |
( ) |
Are the results clearly presented? |
(x) |
( ) |
( ) |
( ) |
Are the conclusions supported by the results? |
(x) |
( ) |
( ) |
( ) |
Comments and Suggestions for Authors
This is a well-written paper highlighting the challenges and complexities in managing TB-HIV coinfection.
The analysis is comprehensive and well-conducted, identifying a whole range of factors that contribute to poorer outcomes.
Would be good to know if authors adjusted or accounted for other comorbidities such as diabetes, end stage renal failure which could also affect outcome.
Comorbidities were accounted for and shown descriptively in the text in the seventh paragraph of the results.
The authors also identify many different factors e.g. social, behavioural, etc which contribute to poorer outcomes. I am sure that these are not specific to Brazil alone. The authors could further strengthen the paper by proposing how they could integrate these key changes into their TB programme, rather than targeting each factor individually, which may be challenging or slow to do.
To improve the poorer results a qualitative study was carried out by our research group and a paragraph was added in the discussion referring to reference number 30. Available at: https://www.mdpi.com/1660-4601/19/22/15153
Submission Date
11 November 2022
Date of this review
20 Jan 2023 07:29:45

Reviewer 4 Report
The topic is important for Brazil, as the country scores among the worst in the world on TB HIV care. However, the study can improve on certain aspects.
Introduction
Line 32,33: HIV remains a challenge; this might be true, but the co-infection rate decreased from 30 to 10 per 100.000 during last 10 years. Also in contrary what is stated here, the mortality rates due to HIV decreased from 11 to 2,5 per 100,000 during the same period . The increase in mortality in TB of the last years , is mainly due to Covid19. The reference used from 2011 is also very old.
Line 38: You may include in this introduction the mortality data from Brazil itself as well.
Line 40: Not clear what is meant with favors
46: How significant is ADR in Brazil?
My suggestion is to focus in the introduction on the essential data around TBHIV ( see title), such as: The extremely low (47%) Treatment success of TBHIV patients; the proportion of TBHIV pts on ART is also very low (50%) , TB pts knowing there HIV status, is low ( 80%) , TPT application is not known. Maybe lack in policies on TB HIV…etc. this to justify the study
Method.
63: The study is hospital based; these are probably very ill patients and not a representative cohort. Later in the table I learn that 19 patients are however non-hospitalized.
72: how is diagnosis ascertained? Culture? Xpert? This needs described, and needs to be for the entire study population the same.
89-95: Reading the types of regimens suggest that some treatments were provided to MDR patients, However MDR is not mentioned. For a study like this, it is extremely important to stratify the study population by the two most important groups: Drug susceptible TB (DS-TB) and drug resistance TB ( DR-TB). It is known that treatment success among MDR patients is lower as compared to patients with sensitive strains, but it looks as if all patients in this study are mixed together. My suggestion is to include only DS-TB patients in the study population or al least make analysis for the two different group DS-TB and DR-TB..
102-122 I suggest not to include these definitions in the article, it is common knowledge, and the references are fine. Moreover, ‘End of treatment” should read ‘Treatment completed’.
130: the analysis requires a sample of 160 patients, while actually 244 patients have been included in the study population. Another important weakness of this study is the fact that of these 244 patients, only 102 (42%) has a known treatment outcome and 142 (58%) are censored and as such has no known treatment outcome. Later in the article (line 174-176), there is more information suggesting that actually from 45,9% of the patients (29,9% transferred out,+ 11.1 % abandoned ,+ 4.9% still undertreatment) treatment outcome is not known. Still very low. Thus from only 54,1% of the 244 patients treatment outcome is known, corresponding with 132 patients. These group should be the study population, which is however lower that the 160 required and also need to be stratified by DS and DR TB, and may provide more convincing study results. _
Results:
In the tables I would suggest three improvements: 1) include besides cure, columns with ‘treatment completed’ and ‘died’. The column censored makes only sense if the umbers are very small, but preferably it should not be included ( see comment under methods). 2) The tables would also improve in clarity when including the percentages of the treatment outcomes. And 3) it is not clear to me what is meant with ‘Median survival days’; Is indeed 50% of the 192 males dead after 227 days?. I can hardly belief. Pls make it clearer or delete the column.
For me the most striking outcome is the low number of women among the patients, but also the extremely low (19%) cure rate among them; what is going on here? Another one is the number of patients not hospitalized (19) (I thought all patients were hospitalised as written in the intro), but particularly the significant high cure rate among this small group (95%). Why can such results ot achieved among males?
190. The basic regimen has better outcomes that the special regimen; yes but, besides there is so much unknown on the treatment outcomes, this is probably a result of the type of patients ( MDR) rather than the regimen itself.
discussion :
216: add that it is a challenge in Brazil. Factors for good treatment outcomes such as being a female and being not hospitalised, require also discussion.
A main reason, why the outcomes on successful treatment are so low, is the fact that the rates of transferred out and abandoned are so high. These are administrative causes and need to be discussed. Issues around basic HIV TB care, as mentioned earlier, require also discussion.
310 A major limitations is the high rate of unknown treatment outcomes as well as the lack of stratification on sensitive ad MDR
Conclusion: early diagnosis and timely treatment favor greater survival: this is true and actually common knowledge, but this is not a conclusion of this study.
Recommendation: My suggestion is to re-analyse the data on study population that includes only DS-TB patients or al least stratify by DS and DR , and only include patients with a reported treatment outcome.
Author Response
Reviewer 4
Open Review
( ) I would not like to sign my review report
(x) I would like to sign my review report
English language and style
( ) English very difficult to understand/incomprehensible
( ) Extensive editing of English language and style required
(x) Moderate English changes required
( ) English language and style are fine/minor spell check required
( ) I don't feel qualified to judge about the English language and style
Yes |
Can be improved |
Must be improved |
Not applicable |
|
Does the introduction provide sufficient background and include all relevant references? |
( ) |
( ) |
(x) |
( ) |
Are all the cited references relevant to the research? |
( ) |
(x) |
( ) |
( ) |
Is the research design appropriate? |
( ) |
( ) |
(x) |
( ) |
Are the methods adequately described? |
( ) |
( ) |
(x) |
( ) |
Are the results clearly presented? |
( ) |
( ) |
(x) |
( ) |
Are the conclusions supported by the results? |
( ) |
( ) |
(x) |
( ) |
Comments and Suggestions for Authors
The topic is important for Brazil, as the country scores among the worst in the world on TB HIV care. However, the study can improve on certain aspects.
Introduction
Line 32,33: HIV remains a challenge; this might be true, but the co-infection rate decreased from 30 to 10 per 100.000 during last 10 years. Also in contrary what is stated here, the mortality rates due to HIV decreased from 11 to 2,5 per 100,000 during the same period . The increase in mortality in TB of the last years , is mainly due to Covid19. The reference used from 2011 is also very old.
As described in the text “Many advances on tuberculosis control in Brazil were achieved in Brazil”. More current references have been added that address this aspect also in the world. The 2011 reference was replaced by a more current reference (2022) that addresses the challenge of controlling TB and HIV/AIDS coinfection.
Line 38: You may include in this introduction the mortality data from Brazil itself as well.
Information on Brazilian mortality was included as suggested.
Line 40: Not clear what is meant with favors
According to the questioning we replaced the word favors with increases and changed the sentence for better understanding.
46: How significant is ADR in Brazil?
Adverse drug reactions interfere with the treatment of coinfected patients, as they can cause unpleasant effects such as hepatotoxicity, nephrotoxicity, psychosis, hypersensitivity, thrombocytopenia, hematological alterations, and rhabdomyolysis, as described in the text. Major reactions may require switching pharmacotherapeutic regimens.
My suggestion is to focus in the introduction on the essential data around TBHIV ( see title), such as: The extremely low (47%) Treatment success of TBHIV patients; the proportion of TBHIV pts on ART is also very low (50%) , TB pts knowing there HIV status, is low ( 80%) , TPT application is not known. Maybe lack in policies on TB HIV…etc. this to justify the study
Text was modified as suggested and added the needs for policies to control TB and HIV/AIDS.
Method.
63: The study is hospital based; these are probably very ill patients and not a representative cohort. Later in the table I learn that 19 patients are however non-hospitalized.
Eduardo de Menezes Hospital (HEM), of the Hospital Foundation of the State of Minas Gerais (FHEMIG), a reference for TB, HIV/AIDS, and other infectious diseases, serves outpatients and inpatients. All patients were followed during the treatment of tuberculosis in the hospital, in the outpatient clinic and in the day hospital. This information was added in the methodology
The 19 patients did not require hospitalization during TB treatment as they were monitored at outpatient clinic.
72: how is diagnosis ascertained? Culture? Xpert? This needs described, and needs to be for the entire study population the same.
The diagnosis of TB and HIV/AIDS was carried out through the evaluation of a specialist physician. For TB signs and symptoms, results of laboratory tests such as detection of acid-fast bacilli in sputum smears or positive culture of a sputum sample or other sterile site were evaluated. And for HIV/AIDS, the diagnosis was made through laboratory tests and rapid tests which detect antibodies against HIV. This excerpt was described in the text, as suggested.
89-95: Reading the types of regimens suggest that some treatments were provided to MDR patients, However MDR is not mentioned. For a study like this, it is extremely important to stratify the study population by the two most important groups: Drug susceptible TB (DS-TB) and drug resistance TB ( DR-TB). It is known that treatment success among MDR patients is lower as compared to patients with sensitive strains, but it looks as if all patients in this study are mixed together. My suggestion is to include only DS-TB patients in the study population or al least make analysis for the two different group DS-TB and DR-TB.
It is not possible to stratify as the number of DR-TB patients is very small as there were only four patients.
We recognized in the discussion that the inclusion of drug resistance TB patients in the study represents a limitation, but because there were only a few patients, we did not change the analysis.
102-122 I suggest not to include these definitions in the article, it is common knowledge, and the references are fine. Moreover, ‘End of treatment” should read ‘Treatment completed’.
We chose to include the definition of outcomes because it is the focus of the study and facilitates the reader's understanding, in addition to emphasizing that the event evaluated in the survival analysis was the success of the TB treatment. The term ‘End of treatment’ has been replaced by ‘Treatment completed’ as suggested.
130: the analysis requires a sample of 160 patients, while actually 244 patients have been included in the study population. Another important weakness of this study is the fact that of these 244 patients, only 102 (42%) has a known treatment outcome and 142 (58%) are censored and as such has no known treatment outcome. Later in the article (line 174-176), there is more information suggesting that actually from 45,9% of the patients (29,9% transferred out,+ 11.1 % abandoned ,+ 4.9% still undertreatment) treatment outcome is not known. Still very low. Thus from only 54,1% of the 244 patients treatment outcome is known, corresponding with 132 patients. These group should be the study population, which is however lower that the 160 required and also need to be stratified by DS and DR TB, and may provide more convincing study results. _
This study considers the real outcome of patients diagnosed with TB and coinfected with HIV/AIDS from 2015 to 2019. During follow-up in our study, it was observed that this population is difficult to follow, considering the high rate of abandonment, death and transfer. The follow-up time was one year and the outcome was evaluated at the end of this period for all patients, as described in the methodology. If only patients with known treatment results were considered, there would be a bias in the selection, since all 102 patients were cured or treatment finished. As reported in the previous questions, it is not practicable to stratify by DS and DR as the number of DR patients is very small (n=4).
Results:
In the tables I would suggest three improvements: 1) include besides cure, columns with ‘treatment completed’ and ‘died’. The column censored makes only sense if the umbers are very small, but preferably it should not be included ( see comment under methods).
The censored column was removed as suggested. Cured and treatment completed was considered as the effectiveness of the treatment (success), as described in the method definitions.
2) The tables would also improve in clarity when including the percentages of the treatment outcomes.
Percentages of treatment outcomes were included as suggested.
3) it is not clear to me what is meant with ‘Median survival days’; Is indeed 50% of the 192 males dead after 227 days?. I can hardly belief. Pls make it clearer or delete the column.
The column has been deleted as suggested. The evaluated outcome was the effectiveness (success) of TB treatment, defined as cure or completion of treatment. The time in the table described was the time in which 50% of the patients survived. The time for healing is longer for females and for patients who required hospitalization during TB follow-up. This information was added in results.
For me the most striking outcome is the low number of women among the patients, but also the extremely low (19%) cure rate among them; what is going on here? Another one is the number of patients not hospitalized (19) (I thought all patients were hospitalised as written in the intro), but particularly the significant high cure rate among this small group (95%). Why can such results ot achieved among males?
Of the total number of women, only 10 were cured. Percentages have been added to the table to improve data visualization. Showing that women cure less than men, considering the proportion of patients.
- The basic regimen has better outcomes that the special regimen; yes but, besides there is so much unknown on the treatment outcomes, this is probably a result of the type of patients ( MDR) rather than the regimen itself.
There are only four MDR patients, which can interfere with the type of scheme. We agree with your suggestion and this point was further discussed and recognized as a limitation of the study.
discussion :
216: add that it is a challenge in Brazil. Factors for good treatment outcomes such as being a female and being not hospitalised, require also discussion.
Factors such as gender and hospitalization were added to the discussion, as suggested.
A main reason, why the outcomes on successful treatment are so low, is the fact that the rates of transferred out and abandoned are so high. These are administrative causes and need to be discussed. Issues around basic HIV TB care, as mentioned earlier, require also discussion.
It was included in the introduction that to achieve successful treatment goals, policies for the control of TB and HIV/AIDS are needed. Furthermore, in the discussion the following excerpt was added: “The main reason why successful treatment outcomes are so low is the fact that transfer and dropout rates are so high. These are administrative causes and need to be better elucidated through the development of public policies and new qualitative studies that assess the social issues in which patients are inserted, such as the use of illicit drugs and the experience with the use of multiple medications in co-infection. ” Reference number 30, a qualitative study that evaluated the subjective experience with the use of medication and its relationship with unfavorable outcomes, was added.
310 A major limitations is the high rate of unknown treatment outcomes as well as the lack of stratification on sensitive ad MDR
These limitations have been added in the discussion.
Conclusion: early diagnosis and timely treatment favor greater survival: this is true and actually common knowledge, but this is not a conclusion of this study.
The conclusion has been rewritten. TB patients coinfected with HIV/ was not effective. This study identified that an undetectable viral load at the beginning of the disease, previous use of ART, not using illicit drugs and not having previously taken anti-TB treatment are factors associated with successful TB treatment.
Recommendation: My suggestion is to re-analyse the data on study population that includes only DS-TB patients or al least stratify by DS and DR , and only include patients with a reported treatment outcome.
Stratification would not be possible, as the group of DR patients is very small, totaling four. We recognized in the discussion that the inclusion of these patients in the study represents a limitation, but because there were only a few patients, we did not change the analysis.
Submission Date
11 November 2022
Date of this review
06 Feb 2023 10:49:58

Round 2
Reviewer 2 Report
In the previous review I pointed out that the Kaplan-Meier survival curves shown in Figure 2 indicated that the use of illicit drugs and lack of virologic suppression at the onset of tuberculosis treatment were associated with longer survival, which is opposite the conclusion of this report. The authors did not address this in their response, and that figure was removed from their revised mss. Based on this inconsistency I could not recommend publication of this paper.
Reviewer 4 Report
the authors replied thouroughly on the comments and accomodated most f the recommedations to improve the article. The fact that many patients have an unkown treatment outcome (abandon and transferred out) remains a weak point of the study, but has been addressed as being adminiatrive issues and clearer mentioned as a limitation.
I have one thing to be clarified: the authors mention "Comparing the time to cure according to the Log-Rank test of the Kaplan-Meier analysis shows a significant difference for sex, illicit drug, hospitalization, and viral suppression (Table 1). The time for healing for females and for patients who required hospitalization during TB follow-up is longer" . My point here is, that is not indicated in the table that time for healing is longer: are, for example women longer treated than the standard 6 months? were the average hospitalization days longer? (is so, good to mention), was the sputum conversion in women later than among men? May be the term time for healing need better described or something else is meant. May be treatment succes rate, but this has little to do with time to healing. Pls clarify